# Optimization over Sparse Support-Preserving Sets via Two-Step Projection

## Abstract

In sparse optimization, enforcing hard constraints using the $\ell_0$ pseudo-norm offers advantages like controlled sparsity compared to convex relaxations. However, many real-world applications (e.g., portfolio optimization) demand not only sparsity constraints but also some extra constraint (such as limit of budget). While prior algorithms have been developed to address this complex scenario with mixed combinatorial and convex constraints, they typically require the closed form projection onto the mixed constraints which might not exist, and/or only provide local guarantees of convergence which is different from the global guarantees commonly sought in sparse optimization. To fill this gap, in this paper, we study the problem of sparse optimization with extra *support-preserving* constraints commonly encountered in the literature. We present a new variant of iterative hard-thresholding algorithm equipped with a two-step consecutive projection operator customized for these mixed constraints, serving as a simple alternative to the Euclidean projection onto the mixed constraint. By introducing a novel trade-off between sparsity relaxation and sub-optimality, we provide global guarantees in objective value for the output of our algorithm, in the deterministic, stochastic, and zeroth-order settings, under the conventional restricted strong-convexity/smoothness assumptions. As a fundamental contribution in proof techniques, we develop a novel extension of the classic three-point lemma to the considered two-step non-convex projection operator, which allows us to analyze the convergence in objective value in an elegant way that has not been possible with existing techniques. Finally, we illustrate the applicability of our method on several sparse learning tasks.

## 1 Introduction

In sparse optimization, directly enforcing sparsity with the $\ell_0$ pseudo-norm has several advantages over its convex relaxation counterpart. In compressive sensing for instance (Foucart & Rauhut, 2013), one may seek to recover an unknown vector, which sparsity level is known to be at most $k$. Similarly, in portfolio optimization, due to transaction costs, one may seek to ensure hard constraints on the maximum number of assets invested in (Brodie et al., 2009; DeMiguel et al., 2009). However, in several use cases, one may also seek to enforce additional constraints, such as, for instance, a budget constraint in the case of portfolio optimization, which can be enforced through an extra $\ell_1$ constraint, as in Takeda et al. (2013). As another example, in sparse non-negative matrix factorization, when estimating the hidden components, one seeks to enforce at the same time a norm constraint and a sparsity constraint Hoyer (2002). The problem of $\ell_0$ empirical risk minimization (ERM) with additional constraints can be formulated as follows, where $R$ is an empirical risk function, $\Gamma \subseteq \mathbb{R}^d$ denotes a convex constraint set, and $\| \cdot \|_0$ denotes the $\ell_0$ pseudo-norm (number of non-zero components of a vector):

$$\min_{\boldsymbol{w} \in \mathbb{R}^p} R(\boldsymbol{w}), \ \text{ s.t. } \|\boldsymbol{w}\|_0 \le k \text{ and } \boldsymbol{w} \in \Gamma. \tag{1}$$

In the literature, several algorithms have been developed to address such a problem with mixed constraints, but they typically require the existence of a closed form for the projection onto the mixed constraint, and/or their convergence guarantees are only local, which makes it difficult to estimate the sub-optimality of the output of the algorithm. More precisely, on one hand, some works provide convergence analyses for variants of a (non-convex) projected gradient descent, explicitly for mixed

sparse constraints (Metel, 2023; Pan et al., 2017; Lu, 2015; Beck & Hallak, 2016), or for general proximal terms (which encompasses our mixed constraints) (Frankel et al. (2014); Xu et al. (2019b); Attouch et al. (2013); De Marchi & Themelis (2022); Yang & Yu (2020); Gu et al. (2018); Yang & Li (2023); Bolte et al. (2014); Boţ et al. (2016); Xu et al. (2019a); Li & Lin (2015)), but such analyses are only local. On the other hand, several existing works on Iterative Hard Thresholding (IHT) provide global guarantees on sub-optimality gap (Jain et al., 2014; Nguyen et al., 2017; Li et al., 2016; Shen & Li, 2017; de Vazelhes et al., 2022), but they do not apply to the mixed constraint case we consider. In between the two approaches, one can also find Barber & Ha (2018) andLiu & Foygel Barber (2020) which give global guarantees for general non-convex constraints or projection operators, but which do not provide explicit convergence guarantees for the particular mixed constraint setting that we consider: their rates depend on some constants (the relative concavity or the local concavity constant) for which, up to our knowledge, an explicit form is still unknown for the mixed constraints we consider. We present a more detailed review of related works in Appendix B, and an overview of them in Table 1. To fill this gap, we focus on solving problem 1 in the case where $\Gamma$ belongs to a general family of *support-preserving* sets, which encompasses many usual sets encountered in the literature. As will be described in more detail in Section 2, such sets are convex sets for which the projection of a $k$-sparse vector onto them gets its support preserved, such as for instance $\ell_p$ norm balls (for $p \geq 1$), or a broader family of *sign–free* convex sets described for instance in Lu (2015); Beck & Hallak (2016).

Adapted to the properties of such constraints, we propose a new variant of IHT, with a two-step projection operator, which, as a first step, identifies the set $S$ of coordinates of the top $k$ components of a given vector and sets the other components to 0 (hard-thresholding), and as a second step projects the resulting vector onto $\Gamma$. This two-step projection can offer a simpler alternative to Euclidean projection onto the mixed constraint in the cases where there is a closed form for the latter projection, and handle the cases where there is not. We then provide global sub-optimality guarantees without system error for the objective value, for such an algorithm as well as its stochastic and zeroth-order variants, under the restricted strong-convexity (RSC) and restricted smoothness (RSS) assumptions, in Theorems 1, 2, and 3. Key to our analysis is a novel extension of the three-point lemma to such non-convex setting with mixed constraints, which also allows, as a byproduct, to simplify existing proofs of convergence in objective value for IHT and its variants. In the zeroth-order case, such technique also allows to obtain, up to our knowledge, the first convergence in risk result without system error for a zeroth-order hard-thresholding algorithm. Additionally, our results highlight a compromise between sparsity and sub-optimality gap specific to the additional constraints setting: through a free parameter $\rho$, one can obtain smaller upper bounds in terms of risk but at the cost of relaxing further the sparsity level of the iterates, or, alternatively, enforce sparser iterates but at the cost of a larger upper bound on the risk.

Finally, we illustrate the applicability of our method on several sparse learning tasks, namely index tracking for portfolio selection, multiclass logistic regression, and adversarial attacks.

**Contributions:** We summarize the main contributions of our paper as follows:

1. We present a variant of IHT to solve hard sparsity problems with additional support-preserving constraints, using a novel two-step projection operator.

2. We describe a novel extension of the three-point lemma to such constraint which allows to simplify existing proofs for IHT and to provide global convergence guarantees in objective value without system error for the algorithm above, in the RSC/RSS setting, highlighting a novel trade-off between sparsity of iterates and sub-optimality gap in such mixed constraints setting.

3. We extend the above algorithm to the stochastic and zeroth-order optimization settings, obtaining similar global convergence guarantees in objective value (without system error) for such mixed constraints setting. In the zeroth-order case, this also provides, up to our knowledge, the first convergence result in objective value without system error for a zeroth-order hard-thresholding algorithm (with or without extra constraints).

Table 1: Comparison of results for Iterative Hard Thresholding with/without additional constraints.
[1] $\mathcal{S}$: symmetric convex sets being sign-free or non-negative (Lu, 2015), $\mathcal{A}$: sets verifying Assumption 3. [2] If a paper reports both $\|\boldsymbol{w} - \bar{\boldsymbol{w}}\|$ and $R(\boldsymbol{w}) - R(\bar{\boldsymbol{w}})$, we report only the latter. $\hat{T}$: time index of the $\boldsymbol{w}$ returned by the method (e.g. $\hat{T} = \arg\min_{t \in [T]} R(\boldsymbol{w}_t)$ ). $\bar{\boldsymbol{w}}$: $\bar{k}$-sparse vector in $\Gamma$. $\Delta$: System error (term which depends on the gradient at optimality (e.g. $\mathbb{E}_i \|\nabla R_i(\bar{\boldsymbol{w}})\|$, (see corresponding references))). [4]: $\kappa_s = \frac{L_s}{\nu_s}$ and $\kappa_{s'} = \frac{L_{s'}}{\nu_s}$ (cf. corresponding refs. for defs. of $s$ and $s'$). [3] SM: Lipschitz-smooth, D: Deterministic. S: Stochastic, Z: Zeroth-Order, L: Lipschitz continuous.

| Reference | $\Gamma$[1] | Convergence[2] | $k$ | Setting[3] |
|---|---|---|---|---|
| Jain et al. (2014) | $\mathbb{R}^d$ | $R(\boldsymbol{w}_{\hat{T}}) \leq R(\bar{\boldsymbol{w}}) + \varepsilon$ | $\Omega(\kappa_s^2 \bar{k})$ | D, RSS, RSC |
| Nguyen et al. (2017) | $\mathbb{R}^d$ | $\mathbb{E}\|\boldsymbol{w}_{\hat{T}} - \bar{\boldsymbol{w}}\| \leq \varepsilon + \mathcal{O}(\Delta)$ | $\Omega(\kappa_s^2 \bar{k})$ | S, RSS, RSC |
| Li et al. (2016) | $\mathbb{R}^d$ | $\mathbb{E}R(\boldsymbol{w}_{\hat{T}}) \leq R(\bar{\boldsymbol{w}}) + \varepsilon + \mathcal{O}(\Delta)$ | $\Omega(\kappa_s^2 \bar{k})$ | S, RSS, RSC |
| Zhou et al. (2018) | $\mathbb{R}^d$ | $\mathbb{E}R(\boldsymbol{w}_{\hat{T}}) \leq R(\bar{\boldsymbol{w}}) + \varepsilon$ | $\Omega(\kappa_s^2 \bar{k})$ | S, RSS, RSC |
| de Vazelhes et al. (2022) | $\mathbb{R}^d$ | $\mathbb{E}\|\boldsymbol{w}_{\hat{T}} - \bar{\boldsymbol{w}}\| \leq \varepsilon + \mathcal{O}(\Delta) + \mathcal{O}(\mu)$ | $\Omega(\kappa_{s'}^4 \bar{k})$ | S, Z, RSS', RSC |
| Lu (2015), Beck & Hallak (2016) | $\Gamma \in \mathcal{S}$ | local convergence | - | D, SM |
| Metel (2023) | $\ell_\infty$ ball around 0 | local convergence | - | S, Z, L |
| **IHT-TSP** (Thm. 1) | $\Gamma \in \mathcal{A} \supset \mathcal{S}$ | $R(\boldsymbol{w}_{\hat{T}}) \leq (1 + 2\rho)R(\bar{\boldsymbol{w}}) + \varepsilon$ | $\Omega\left(\frac{\kappa_s^2 \bar{k}}{\rho^2}\right)$ | D, RSS, RSC |
| **HSG-HT-TSP** (Thm. 2) | $\Gamma \in \mathcal{A} \supset \mathcal{S}$ | $\mathbb{E}R(\boldsymbol{w}_{\hat{T}}) \leq (1 + 2\rho)R(\bar{\boldsymbol{w}}) + \varepsilon$ | $\Omega\left(\frac{\kappa_s^2 \bar{k}}{\rho^2}\right)$ | S, RSS, RSC |
| **HZO-HT-TSP** (Thm. 3) | $\Gamma \in \mathcal{A} \supset \mathcal{S}$ | $\mathbb{E}R(\boldsymbol{w}_{\hat{T}}) \leq (1 + 2\rho)R(\bar{\boldsymbol{w}}) + \varepsilon + \mathcal{O}(\mu)$ | $\Omega\left(\frac{\kappa_{s'}^2 \bar{k}}{\rho^2}\right)$ | Z, RSS', RSC |
| **HZO-HT** (Thm. 6 in App. E.3.2) | $\mathbb{R}^d$ | $\mathbb{E}[R(\boldsymbol{w}_{\hat{T}}) - R(\bar{\boldsymbol{w}})] \leq \varepsilon + \mathcal{O}(\mu)$ | $\Omega(\kappa_{s'}^2 \bar{k})$ | Z, RSS', RSC |

## 2 PRELIMINARIES

Throughout this paper, we adopt the following notations. For any $\boldsymbol{w} \in \mathbb{R}^d$, $\Pi_\Gamma(\boldsymbol{w})$ denotes a Euclidean projection of $\boldsymbol{w}$ onto $\Gamma$, that is $\Pi_\Gamma(\boldsymbol{w}) \in \arg\min_{\boldsymbol{z} \in \Gamma} \|\boldsymbol{w} - \boldsymbol{z}\|_2$, and $w_i$ denotes the $i$-th component of $\boldsymbol{w}$. $\mathcal{B}_0(k)$ denotes the $\ell_0$ pseudo-ball of radius $k$, i.e. $\mathcal{B}_0(k) = \{\boldsymbol{w} \in \mathbb{R}^d : \|\boldsymbol{w}\|_0 \leq k\}$, with $\|\cdot\|_0$ the $\ell_0$ pseudo-norm (i.e. the number of nonzero components of a vector). $\mathcal{H}_k$ denotes the Euclidean projection onto $\mathcal{B}_0(k)$, also known as the hard-thresholding operator (which keeps the $k$ largest (in magnitude) components of a vector, and sets the others to 0 (if there are ties, we can break them e.g. lexicographically)). $\|\cdot\|_p$ denotes the $\ell_p$ norm for $p \in [1, +\infty)$, and $\|\cdot\|$ the $\ell_2$ norm (unless otherwise specified). $[n]$ denotes the set $\{1, ..., n\}$ for $n \in \mathbb{N}^*$. For any $S \subseteq [d]$, $|S|$ denotes its number of elements. For any $\boldsymbol{w} \in \mathbb{R}^d$, $\text{supp}(\boldsymbol{w})$ denotes its support, i.e. the set of coordinates of its non-zero components. We will also need the following assumptions on $R$.

**Assumption 1** (($\nu_s, s$)-RSC, Jain et al. (2014); Negahban et al. (2009); Loh & Wainwright (2013); Yuan et al. (2017); Li et al. (2016); Shen & Li (2017); Nguyen et al. (2017)). *$R$ is $\nu_s$ restricted strongly convex with sparsity parameter $s$, i.e. it is differentiable, and there exists a generic constant $\nu_s$ such that for all $(\boldsymbol{x}, \boldsymbol{y}) \in \mathbb{R}^d$ with $\|\boldsymbol{x} - \boldsymbol{y}\|_0 \leq s$:*

$$R(\boldsymbol{y}) \geq R(\boldsymbol{x}) + \langle \nabla R(\boldsymbol{x}), \boldsymbol{y} - \boldsymbol{x} \rangle + \frac{\nu_s}{2} \|\boldsymbol{x} - \boldsymbol{y}\|^2$$

**Assumption 2** (($L_s, s$)-RSS, Jain et al. (2014); Li et al. (2016); Yuan et al. (2017)). *$R$ is $L_s$ restricted smooth with sparsity level $s$, i.e. it is differentiable, and there exists a generic constant $L_s$ such that for all $(\boldsymbol{x}, \boldsymbol{y}) \in \mathbb{R}^d$ with $\|\boldsymbol{x} - \boldsymbol{y}\|_0 \leq s$:*

$$R(\boldsymbol{y}) \leq R(\boldsymbol{x}) + \langle \nabla R(\boldsymbol{x}), \boldsymbol{y} - \boldsymbol{x} \rangle + \frac{L_s}{2} \|\boldsymbol{x} - \boldsymbol{y}\|^2$$

We then define the notion of support-preserving set that we will use throughout the paper. It essentially requires that projecting any $k$-sparse vector $\boldsymbol{w}$ onto $\Gamma$ preserves its support. That is, the convex constraint $\Gamma$ should be compatible to the sparsity level constraint $\|\boldsymbol{w}\|_0 \leq k$.

**Assumption 3** ($k$-support-preserving set). *$\Gamma \subseteq \mathbb{R}^d$ is $k$-support-preserving , i.e. it is convex and for any $\boldsymbol{w} \in \mathbb{R}^d$ such that $\|\boldsymbol{w}\|_0 \leq k$, $supp(\Pi_\Gamma(\boldsymbol{w})) \subseteq supp(\boldsymbol{w})$.*

**Remark 1.** *Below we present some examples of usual sets that also verify Assumption 3 (see Appendix C for a proof of such statements):*

- *Elementwise decomposable constraints, such as box constraints of the form $\{\boldsymbol{w} \in \mathbb{R}^d : \forall i \in [d], l_i \leq w_i \leq u_i\}$.*
- *Group-wise separable constraints where the constraint on each group is $k$-support-preserving (such as our constraints in Section 5 for the index tracking problem).*
- *Sign-free convex sets (Lu, 2015; Beck & Hallak, 2016) (def. in App. C), e.g. $\ell_q$ norm-balls.*

## 3 DETERMINISTIC CASE

### 3.1 ALGORITHM

**Two-step projection** In all the algorithms of this paper, we will make use of a *two-step projection* operator (TSP), which is different in general from the usual Euclidean projection (EP), in order to obtain, from an arbitrary vector $\boldsymbol{w} \in \mathbb{R}^d$, a vector in $\boldsymbol{w} \in \mathcal{B}_0(k) \cap \Gamma$. We consider such a TSP instead of EP since it enables the derivation a variant of three-point lemma (Lemma 1) which can handle our specific non-convex mixed constraints, and is key to obtaining the convergence analyses we present in Sections 3 and 4. In addition, the TSP can be more intuitive and efficient to implement than EP (see App. F.2 for more discussions about TSP vs EP). The TSP procedure, which we denote by $\bar{\Pi}_\Gamma^k$, is as follows: we first project $\boldsymbol{w}$ onto $\mathcal{B}_0(k)$ through the hard-thresholding operator $\mathcal{H}_k$, to obtain a $k$-sparse vector $\boldsymbol{v}_k = \mathcal{H}_k(\boldsymbol{w})$. Then, we project $\boldsymbol{v}_k$ onto $\Gamma$ , to obtain a final vector $\boldsymbol{w}_S = \Pi_\Gamma(\boldsymbol{v}_k)$, where $S = supp(\boldsymbol{v}_k)$. Note that consequently, the ob-

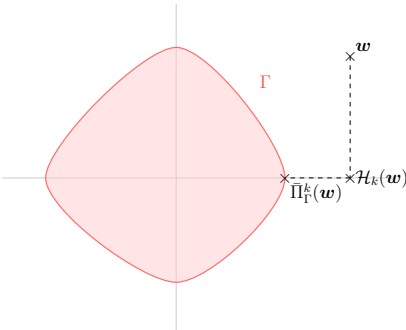

Figure 1: Support-preserving set and two-step projection ($d = 2, k = 1$).

tained $\boldsymbol{w}_S$ is not necessarily the EP of $\boldsymbol{w}$ onto $\mathcal{B}_0(k) \cap \Gamma$, that is, we do not necessarily have $\boldsymbol{w}_S = \Pi_{\mathcal{B}_0(k) \cap \Gamma}(\boldsymbol{w})$. However, when Assumption 3 is verified, we have $\boldsymbol{w}_S \in \mathcal{B}_0(k) \cap \Gamma$ (since, because of Assumption 3, $supp(\boldsymbol{w}_S) \subseteq supp(\boldsymbol{v}_k)$ and hence $\|\boldsymbol{w}_S\|_0 \leq \|\boldsymbol{v}_k\|_0 \leq k$), therefore each iteration remains feasible in the constraint. We illustrate such a two-step projection on Figure 1.

We now present our full algorithm in the case where $R$ is a deterministic function without further knowledge of its structure. It is similar to the usual (non-convex) projected gradient descent algorithm, that is, a gradient update step followed by a projection step, except that instead of projecting onto $\Gamma \cap \mathcal{B}_0(k)$ using the Euclidean projection, we obtain a vector $\boldsymbol{w}_k \in \Gamma \cap \mathcal{B}_0(k)$ through the two-step projection method described above. We describe the algorithm in Algorithm 1 below.

---

**Algorithm 1:** Deterministic IHT with extra constraints (IHT-TSP)

**Input:** $\boldsymbol{w}_0$: initial value, $\eta$: learning rate, $T$: number of iterations
**for** $t = 1$ *to* $T$ **do**
$\quad | \quad \boldsymbol{w}_t \leftarrow \bar{\Pi}_\Gamma^k(\boldsymbol{w}_{t-1} - \eta \nabla R(\boldsymbol{w}_{t-1}))$;
**end**
**Output:** $\boldsymbol{w}_T$

---

**Remark 2.** *In the case where $\Gamma$ is a symmetric sign-free convex set (we refer to Lu (2015) for the definition of such sets, which include for instance any $\ell_p$ norm constraint set for $p \in [1, +\infty)$ ), then the two-step projection is the closed form of an Euclidean projection onto the mixed constraint $\Gamma \cap \mathcal{B}_0(k)$ (see Theorem 2.1 from Lu (2015)). Therefore, in such cases, Algorithm 1 is identical to a vanilla (non-convex) projected gradient descent algorithm (for which up to now there was still no global convergence guarantees in such a mixed constraints setting in the literature).*

## 3.2 CONVERGENCE ANALYSIS

Before proceeding with the convergence analysis, we first present below a variant of the usual three-point lemma, which plays a key role in the proof. The three-point lemma is usually used in the proofs for projected gradient descent in the convex setting. However, due to the non-convexity of the $\ell_0$ pseudo-ball, such proofs cannot apply, and to provide convergence in risk, some complex work-arounds are often taken via careful consideration of the sizes of the support of the iterates, such as in the proofs of Jain et al. (2014) or Zhou et al. (2018). However, such a modified three-point lemma below allows to obtain simpler proofs in such non-convex setting, remaining very close to usual convex optimization proofs, while also being able to take into account the additional constraint, which is important in our problem setting. More specifically, the common three-point lemma for a projection onto a convex set $\mathcal{E}$ relates the distance between a point $\boldsymbol{w} \in \mathbb{R}^d$, its projection $\Pi_{\mathcal{E}}(\boldsymbol{w})$, and any vector $\bar{\boldsymbol{w}}$ from the set $\mathcal{E}$, through the relation $\|\boldsymbol{w} - \bar{\boldsymbol{w}}\|^2 \geq \|\Pi_{\mathcal{E}}(\boldsymbol{w}) - \boldsymbol{w}\|^2 + \|\Pi_{\mathcal{E}}(\boldsymbol{w}) - \bar{\boldsymbol{w}}\|^2$. However, in our case, our lemma relates together the four points involved in the two step projection ($\boldsymbol{w} \in \mathbb{R}^d$, $\mathcal{H}_k(\boldsymbol{w})$, $\bar{\Pi}_{\Gamma}^k(\boldsymbol{w})$, and $\bar{\boldsymbol{w}} \in \Gamma \cap \mathcal{B}_0(k)$ ), and additionally, it contains a constant $\beta$ which takes into account the sparsity level $k$ enforced in the algorithm and the sparsity $\bar{k}$ ($< k$) of a reference point $\bar{\boldsymbol{w}}$ (see e.g. Liu & Foygel Barber (2020) for a discussion regarding $k$ and $\bar{k}$).

**Lemma 1** (Constrained $\ell_0$-Three-Point, proof in App. D.1). *Suppose that Assumption 3 holds. Consider $\boldsymbol{w}, \bar{\boldsymbol{w}} \in \mathbb{R}^p$ with $\|\bar{\boldsymbol{w}}\|_0 \leq \bar{k}$ and $\bar{\boldsymbol{w}} \in \Gamma$. Then the following holds for any $k > \bar{k}$, with $\beta := \frac{\bar{k}}{k}$:*

$$\|\bar{\Pi}_{\Gamma}^k(\boldsymbol{w}) - \boldsymbol{w}\|^2 \leq \|\boldsymbol{w} - \bar{\boldsymbol{w}}\|^2 - \|\bar{\Pi}_{\Gamma}^k(\boldsymbol{w}) - \bar{\boldsymbol{w}}\|^2 + \sqrt{\beta}\|\mathcal{H}_k(\boldsymbol{w}) - \bar{\boldsymbol{w}}\|^2.$$

In the case where $\Gamma = \mathbb{R}^d$, we have $\bar{\Pi}_{\Gamma}^k(\boldsymbol{w}) = \mathcal{H}_k(\boldsymbol{w})$, and we can observe that if $k \gg \bar{k}$, $\beta$ tends to 0, and therefore we approach the usual three-point lemma from convex optimization. This is coherent with the literature on IHT, in which relaxing the sparsity degree (i.e. considering some $k \gg \bar{k}$) is known to make the problem easier to solve (we refer the reader to references in Appendix B.2 for more details). Equipped with such lemma, we can now present the convergence analysis of Algorithm 1 below, using the assumptions from Section 2, and we will describe how the results give rise to a trade-off between the sparsity of the iterates and the tightness of the sub-optimality bound, specific to our mixed constraints setting.

**Theorem 1** (Proof in App. D.2). *Suppose that Assumption 1, 2, and 3 hold, and that $R$ is non-negative (without loss of generality). Let $s = 2k$, $\eta = \frac{1}{L_s}$, and $\bar{\boldsymbol{w}}$ be an arbitrary $\bar{k}$-sparse vector. Let $\rho \in (0, \frac{1}{2}]$ be an arbitrary scalar. Suppose that $k \geq \frac{4(1-\rho)^2 L_s^2}{\rho^2 \nu_s^2} \bar{k}$. Then for any $\varepsilon > 0$, for*

$$T \geq \left\lceil \frac{L_s}{\nu_s} \log\left(\frac{(L_s - \nu_s)\|\boldsymbol{w}_0 - \bar{\boldsymbol{w}}\|^2}{2\varepsilon(1-\rho)}\right) \right\rceil + 1 = \mathcal{O}(\kappa_s \log(\frac{1}{\varepsilon})),$$ *the iterates of IHT-TSP satisfy*

$$\min_{t \in [T]} R(\boldsymbol{w}_t) \leq (1 + 2\rho)R(\bar{\boldsymbol{w}}) + \varepsilon.$$

*Further, if $\bar{\boldsymbol{w}}$ is a global minimizer of $R$ over $\mathcal{B}_0(k) := \{\boldsymbol{w} : \|\boldsymbol{w}\|_0 \leq k\}$, then, with $\rho = 0.5$ in the expressions of $k$ and $T$ above: $\min_{t \in [T]} R(\boldsymbol{w}_t) \leq R(\bar{\boldsymbol{w}}) + \varepsilon$.*

*Proof Sketch.* Our proof starts by deriving a novel convergence proof for IHT in the case where $\Gamma = \mathbb{R}^d$ (Theorem 4 in Appendix), greatly simplifying the one from Jain et al. (2014) (Proof of Thm. 1 in App. B.1), and much closer to usual constrained convex optimization proofs. Using the $L_s$-RSS of $R$ and some algebraic manipulations, and denoting $\boldsymbol{g}_t = \nabla R(\boldsymbol{w}_t)$ and $\boldsymbol{v}_t := \mathcal{H}_k(\boldsymbol{w}_{t-1} - \frac{1}{L_s}\boldsymbol{g}_{t-1})$ ($= \boldsymbol{w}_t$ when $\Gamma = \mathbb{R}^d$), we have:

$$R(\boldsymbol{v}_t) \leq R(\boldsymbol{w}_{t-1}) + \frac{L_s}{2}\|\boldsymbol{v}_t - \boldsymbol{w}_{t-1} + \frac{1}{L_s}\boldsymbol{g}_{t-1}\|^2 - \frac{1}{2L_s}\|\nabla R(\boldsymbol{w}_{t-1})\|^2$$

$$\overset{(a)}{\leq} R(\boldsymbol{w}_{t-1}) + \frac{L_s}{2}\|\bar{\boldsymbol{w}} - \boldsymbol{w}_{t-1} + \frac{1}{L_s}\boldsymbol{g}_{t-1}\|^2 - \frac{L_s}{2}(1 - \sqrt{\beta})\|\boldsymbol{v}_t - \bar{\boldsymbol{w}}\|^2 - \frac{1}{2L_s}\|\nabla R(\boldsymbol{w}_{t-1})\|^2$$

$$\overset{(b)}{\leq} R(\bar{\boldsymbol{w}}) + \frac{L_s - \nu_s}{2}\|\boldsymbol{w}_{t-1} - \bar{\boldsymbol{w}}\|^2 - \frac{L_s}{2}(1 - \sqrt{\beta})\|\boldsymbol{v}_t - \bar{\boldsymbol{w}}\|^2, \tag{2}$$

where in (a) we used our new $\ell_0$-three-point lemma (Lemma 3 in App. D.1.1), and in (b) we used the RSC of $R$ with some rearrangements. At that stage, the proof for Theorem 4 can be concluded with telescopic sum arguments. To obtain the proof for general $\Gamma$ (i.e. Theorem 1), we reiterate the

above process but instead of Lemma 3 we use our more general Lemma 1, adapted to general $\Gamma$ and to our two-step projection technique, to obtain:

$$R(\boldsymbol{w}_t) \leq R(\bar{\boldsymbol{w}}) + \frac{L_s - \nu_s}{2}\|\boldsymbol{w}_{t-1} - \bar{\boldsymbol{w}}\|^2 - \frac{L_s}{2}\|\boldsymbol{w}_t - \bar{\boldsymbol{w}}\|^2 + \frac{L_s}{2}\sqrt{\beta}\|\boldsymbol{v}_t - \bar{\boldsymbol{w}}\|^2. \qquad (3)$$

Finally, taking a convex combination of equations 2 ($\times \rho$) and 3 ($\times(1-\rho)$) for $\rho \in (0, 0.5]$, using the bound $\|\boldsymbol{w}_t - \bar{\boldsymbol{w}}\|^2 \leq \|\boldsymbol{v}_t - \bar{\boldsymbol{w}}\|^2$ (non-expansiveness of convex projection onto $\Gamma$), and carefully tuning $k$ depending on $\rho$ (resulting in our final trade-off between sparsity and optimality), we can fall back to a telescopic sum and conclude the proof. $\qquad \square$

**Remark 3.** *Theorem 1 therefore provides a global convergence guarantee in objective value. However, contrary to usual guarantees for IHT algorithms under RSS/RSC conditions (which are bounds of the form $R(\boldsymbol{w}_t) \leq R(\bar{\boldsymbol{w}}) + \varepsilon$ for some $t$), our bound is of the form $R(\boldsymbol{w}_t) \leq (1 + 2\rho)R(\bar{\boldsymbol{w}}) + \varepsilon$. There is a trade-off about the choice of $\rho \in (0, 0.5]$. On one hand, $\rho \to 0$ is preferred in view of the RHS of above bound. On the other hand, the sparsity-level relaxation condition $k \geq \frac{4(1-\rho)^2 L_s^2}{\rho^2 \nu_s^2}\bar{k}$ prefers $\rho \to 0.5$. We illustrate such a trade-off on some synthetic experiments in Section F.5.*

## 4 Extensions: Stochastic and Zeroth-Order cases

In this section, we provide extensions of Algorithm 1 to the stochastic and zeroth-order sparse optimization problems, and provide the corresponding convergence guarantees in objective value without system error.

### 4.1 Stochastic Optimization

In this section, we consider the previous risk minimization problem, in a finite-sum setting, i.e. where $R(\boldsymbol{w}) = \frac{1}{n}\sum_{i=1}^{n} R_i(\boldsymbol{w})$, similarly to Zhou et al. (2018); Nguyen et al. (2017): in such case, stochastic algorithms allow to deal more easily with large-scale datasets where estimating the full $\nabla R(\boldsymbol{w})$ is expensive.

#### 4.1.1 Algorithm

We describe the stochastic variant of our previous Algorithm 1 in Algorithm 2 below, which is an extension of the algorithm from Zhou et al. (2018), to the considered mixed constraints problem setting, using our two-step projection. More precisely, we approximate the gradient of $R$ by a mini-batch stochastic gradient with a batch-size increasing exponentially along training, and following the gradient step, we apply our two-step projection operator.

---

**Algorithm 2:** Hybrid Stochastic IHT with Extra Constraints (HSG-HT-TSP)

**Input:** $\boldsymbol{w}_0$: initial point, $\eta$: learning rate, $T$: number of iterations, $\{s_t\}$: mini-batch sizes.
**for** $t = 1$ *to* $T$ **do**
  Uniformly sample $s_t$ indices $\mathcal{S}_t$ from $[n]$ without replacement ;
  Compute the approximate gradient $\boldsymbol{g}_{t-1} = \frac{1}{s_{t-1}}\sum_{i_t \in \mathcal{S}_t} \nabla R_{i_t}(\boldsymbol{w}_{t-1})$
  $\boldsymbol{w}_t = \bar{\Pi}_\Gamma^k(\boldsymbol{w}_{t-1} - \eta\boldsymbol{g}_{t-1})$;
**end**
**Output:** $\hat{\boldsymbol{w}}_T = \arg\min_{\boldsymbol{w} \in \{\boldsymbol{w}_1, \dots, \boldsymbol{w}_T\}} R(\boldsymbol{w})$.

---

#### 4.1.2 Convergence Analysis

Before proceeding with the convergence analysis, we make an additional assumption on the population variance of the stochastic gradients, similar to the one in Mishchenko et al. (2020).

**Assumption 4** (Bounded stochastic gradient variance)**.** *For any $\boldsymbol{w}$, the population variance of the gradient estimator is bounded by $B$:*

$$\frac{1}{n}\sum_{i=1}^{n} \|\nabla R_i(\boldsymbol{w}) - \nabla R(\boldsymbol{w})\|^2 \leq B.$$

We now present our convergence analysis below:

**Theorem 2** (Proof in App. E.1). *Suppose that Assumptions 1 2, 3 and 4 hold, and that R is non-negative (without loss of generality). Let $s = 2k$. Let $\bar{w}$ be an arbitrary $\bar{k}$-sparse vector. Let $C$ be an arbitrary positive constant. Assume that we run HSG-HT-TSP (Algorithm 2) for $T$ timesteps, with $\eta = \frac{1}{L_s + C}$, and denote $\alpha := \frac{C}{L_s} + 1$ and $\kappa_s := \frac{L_s}{\nu_s}$. Suppose that $k \geq 4\alpha^2 \frac{1}{\rho^2} \kappa_s^2 \bar{k}$ for some $\rho \in (0, 1)$. Finally, assume that we take the following batch-size: $s_t := \left\lceil \frac{\tau}{\omega^t} \right\rceil$ with $\omega := 1 - \frac{1}{4\alpha \frac{1}{\rho} \kappa_s}$ and $\tau := \frac{\eta B}{C}$. Then, we have the following convergence rate:*

$$\mathbb{E} \min_{t \in [T]} R\left(w_t\right) - (1 + 2\rho)R(\bar{w}) \leq 2 \frac{\alpha^2}{\rho(1-\rho)} L_s \kappa_s \omega^T \left( \|\bar{w} - w_0\|^2 + \frac{4}{3} \right).$$

*Further, if $\bar{w}$ is a global minimizer of $R$ over $\mathcal{B}_0(k) := \{w : \|w\|_0 \leq k\}$, then, with $\rho = 0.5$:*

$$\mathbb{E} \min_{t \in [T]} R\left(w_t\right) - R(\bar{w}) \leq 8\alpha^2 L_s \kappa_s \omega^T \left( \|\bar{w} - w_0\|^2 + \frac{4}{3} \right).$$

**Corollary 1** (Proof in App. E.2.). *Therefore, the number of calls to a gradient $\nabla R_i$ (#IFO), and the number of hard thresholding operations (#HT) such that the left-hand sides in Theorem 2 above are smaller than some $\varepsilon > 0$, are respectively: #HT $= \mathcal{O}(\kappa_s \log(\frac{1}{\varepsilon}))$ and #IFO $= \mathcal{O}\left( \frac{\kappa_s}{\nu_s \varepsilon} \right)$.*

### 4.2 ZEROTH-ORDER OPTIMIZATION

We now consider the zeroth-order (ZO) case (Nesterov & Spokoiny, 2017), in which one does not have access to the gradient $\nabla R(w)$, but only to function values $R(w)$, which arises for instance when the dataset is private as in distributed learning (Gratton et al., 2021; Zhang et al., 2021) or the model is private as in black-box adversarial attacks Liu et al. (2018), or when computing $\nabla R(w)$ is too expensive such as in certain graphical modeling tasks Wainwright et al. (2008). The idea is then to approximate $\nabla R(w)$ using finite differences. We refer the reader to Berahas et al. (2021) and Liu et al. (2020) for an overview of ZO methods.

#### 4.2.1 ALGORITHM

In this section, we describe the ZO version of our algorithm. At its core, it uses the ZO estimator from de Vazelhes et al. (2022). We present the full algorithm in Algorithm 3, where $\mathcal{D}_{s_2}$ is a uniform probability distribution on the following set $\mathcal{B}$, which is the set of unit spheres supported on supports of size $s_2 \leq d$: $\mathcal{B} = \{w \in \mathbb{R}^d : \|w\|_0 \leq s_2, \|w\|_2 \leq 1\}$. We can sample from this set by first sampling a random support of size $s_2$, and then sampling from the unit sphere on that support. Note that if we choose $s_2 := d$, this estimator simply becomes the vanilla ZO estimator with unit-sphere smoothing (Liu et al., 2020). Choosing $s_2 < d$ allows to avoid the full-smoothness assumption and can reduce memory consumption by allowing to sample random vectors of size $s_2$ instead of $d$. We refer to de Vazelhes et al. (2022) for more details on such a ZO estimator. The difference with de Vazelhes et al. (2022) (in addition to the mixed constraint setting and the use of the TSP) is that in our case we sample an exponentially increasing number of random directions, which allows us to obtain convergence in risk without system error (except the system error due to the smoothing $\mu$).

---

**Algorithm 3:** Hybrid ZO IHT with Extra Constraints (HZO-HT-TSP)

**Input:** $w_0$: initial point, $\eta$: learning rate, $T$: number of iterations, $s_2$: size of the random
        supports, $\{q_t\}$: number of random directions.
**for** $t = 1$ *to* $T$ **do**
    Uniformly sample $q_{t-1}$ i.i.d. random directions $\{u_i\}_{i=1}^{q_{t-1}} \sim \mathcal{D}_{s_2}$ ;
    Compute the approximate gradient $g_t = \frac{1}{q_{t-1}} \sum_{i=1}^{q_{t-1}} \frac{d}{\mu} \left( R(w_{t-1} + \mu u_i) - R(w_{t-1}) \right) u_i$
    $w_t = \bar{\Pi}_\Gamma^k(w_{t-1} - \eta g_{t-1})$;
**end**
**Output:** $\hat{w}_T = \arg\min_{w \in \{w_1, \ldots, w_T\}} R(w)$.

---

### 4.2.2 CONVERGENCE ANALYSIS

**Assumption 5** (($L_s, s$)-RSS', Shen & Li (2017); Nguyen et al. (2017)). *$R$ is $L_s$ strongly restricted smooth with sparsity level $s$, i.e. it is differentiable, and there exist a generic constant $L_s$ such that for all $(\boldsymbol{x}, \boldsymbol{y}) \in \mathbb{R}^d$ with $\|\boldsymbol{x} - \boldsymbol{y}\|_0 \leq s$:*

$$\|\nabla R(\boldsymbol{x}) - \nabla R(\boldsymbol{y})\| \leq L_s \|\boldsymbol{x} - \boldsymbol{y}\|.$$

*Note that if a function $R$ is ($L_s, s$)-RSS', then it is ($L_s, s$)-RSS.*

Such assumption is often simply called restricted smoothness, but we name it strong restricted smoothess to avoid any confusion with Assumption 2. Assumption 5 is slightly more restrictive than Assumption 2, but it is necessary when working with ZO gradient estimators (see more details in de Vazelhes et al. (2022)). We now present our main convergence theorem for the ZO setting.

**Theorem 3** (Proof in App. E.3). *Suppose that Assumptions 1, 3, and 5 hold, and that $R$ is non-negative (without loss of generality). Let $s = 3k$, and let $\bar{\boldsymbol{w}}$ be an arbitrary $\bar{k}$-sparse vector. Let $s_2 \in \{1, ..., d\}$. Assume that $R$ is ($L_{s'}, s'$)-RSS' with $s' = \max(s_2, s)$, and $\nu_s$-RSC. Denote $\kappa_s := \frac{L_{s'}}{\nu_s}$. Let $C$ be an arbitrary positive constant, and denote $\varepsilon_F := \frac{2d}{(s_2+2)} \left( \frac{(s-1)(s_2-1)}{d-1} + 3 \right)$, $\varepsilon_{abs} := 2dL_{s'}^2 s s_2 \left( \frac{(s-1)(s_2-1)}{d-1} + 1 \right)$, and $\varepsilon_\mu := L_{s'}^2 sd$. Assume that we run HZO-HT-TSP (Algorithm 3) for $T$ timesteps, with $\eta = \frac{1}{L_{s'}+C} = \frac{1}{\alpha L_{s'}}$, with $\alpha := \frac{C}{L_{s'}} + 1$. Suppose that $k \geq 16\frac{\alpha^2}{\rho^2}\kappa_s^2 \bar{k}$ for some $\rho \in (0, 1)$. Finally, assume that we take $q_t$ random directions at each iteration, with $q_t := \left\lceil \frac{\tau}{\omega^t} \right\rceil$ with $\omega := 1 - \frac{1}{8\frac{1}{\rho}\alpha\kappa_s}$ and $\tau := 16\kappa_s \frac{\varepsilon_F}{(\alpha-1)}$. Then, we have the following convergence rate:*

$$\mathbb{E} \min_{t \in [T]} R(\boldsymbol{w}_t) - (1 + 2\rho)R(\bar{\boldsymbol{w}}) \leq 4\frac{\alpha^2}{\rho(1-\rho)}L_{s'}\kappa_s\omega^T \left( \|\bar{\boldsymbol{w}} - \boldsymbol{w}_0\|^2 + \frac{1}{3}\frac{\eta\|\nabla R(\bar{\boldsymbol{w}})\|^2}{\kappa_s L_{s'}} \right) + Z\mu^2,$$

*with $Z = \frac{1}{1-\rho} \left( \varepsilon_\mu \left( \frac{2}{\nu_s} + \frac{1}{C} \right) + \frac{\varepsilon_{abs}}{C} \right)$. Further, if $\bar{\boldsymbol{w}}$ is a global minimizer of $R$ over $\mathcal{B}_0(k) := \{\boldsymbol{w} : \|\boldsymbol{w}\|_0 \leq k\}$, then, with $\rho = 0.5$:*

$$\mathbb{E} \min_{t \in [T]} R(\boldsymbol{w}_t) - R(\bar{\boldsymbol{w}}) \leq 16\alpha^2 L_{s'}\kappa_s\omega^T \left( \|\bar{\boldsymbol{w}} - \boldsymbol{w}_0\|^2 + \frac{1}{3}\frac{\eta\|\nabla R(\bar{\boldsymbol{w}})\|^2}{\kappa_s L_{s'}} \right) + Z\mu^2.$$

**Corollary 2** (Proof in App. E.4.). *Additionally, the number of calls to the function $R$ (#IZO), and the number of hard thresholding operations (#HT) such that the left-hand sides in Theorem 3 above are smaller than $\varepsilon + Z\mu^2$, for some $\varepsilon > 0$ are respectively: #HT $= \mathcal{O}(\kappa_s \log(\frac{1}{\varepsilon}))$ and #IZO $= \mathcal{O}\left( \varepsilon_F \frac{\kappa_s^3 L_s}{\varepsilon} \right)$. Note that if $s_2 = d$, we have $\varepsilon_F = \mathcal{O}(s) = \mathcal{O}(k)$, and therefore we obtain a query complexity that is dimension independent.*

**Remark 4.** *If $\Gamma = \mathbb{R}^d$, we name the corresponding algorithm HZO-HT, and we provide the convergence rate of HZO-HT in Theorem 6 in Appendix E.3.2, also recalled in Table 1. Such a result is novel, and can be seen as an independent contribution illustrating the power of proof techniques based on our three-point lemma. Up to our knowledge, it is the first global convergence guarantee without system error for a zeroth-order hard-thresholding algorithm (see Table 1), and as such, is a significant improvement over the result from de Vazelhes et al. (2022).*

## 5 EXPERIMENTS

Before describing our experiments, we provide a short discussion about the settings and algorithms that we will illustrate. For constraints $\Gamma$ for which the Euclidean projection onto $\mathcal{B}_0(k) \cap \Gamma$ has a closed form equal to the TSP, our algorithm is identical to a vanilla non-convex projected gradient descent baseline (see Remark 2). In such case, our contribution in this paper is on the theoretical side, by providing some global guarantees on the optimization, instead of the local guarantees from existing work (cf. Table 1). Additionally, there are case in which there exists a closed form for projection onto $\Gamma \cap \mathcal{B}_0(k)$, different from the TSP (e.g. when $\Gamma = \mathbb{R}_+^d$, cf. Lu (2015)). Although our framework allows us to get approximate global convergence results when using the TSP, still, at the iteration level, a gradient step followed by Euclidean projection (not TSP) is optimal, since it

minimizes a constrained quadratic upper bound on $R$. Therefore, we may not expect much improvement of the TSP over the Euclidean projection in such case, except on the computational side . For these reasons, we illustrate cases where, up to our knowledge, there is no known closed form for projection onto $\Gamma \cap \mathcal{B}_0(k)$, which we believe are the most interesting from the empirical perspective (since no algorithm was about to deal with such cases before). We present below an experiment on a real life index tracking use-case, and provide some extra experimental results in Appendix F, for the settings of multi-class logistic regression as well as adversarial attacks.

**Setting: Index tracking.** We consider the following index tracking problem, originally presented in Takeda et al. (2013), and used as well in Lu (2015); Beck & Hallak (2016). It is also similar to the portfolio optimization problem presented in Kyrillidis et al. (2013). We seek to reproduce the performance of an index fund (such as S&P500), by investing only in a few key $k$ assets, in order to limit transaction costs. The general problem can be formulated as a linear regression problem:

$$\min_{\boldsymbol{w} \in \mathcal{B}_0(k) \cap \Gamma} \|\boldsymbol{A}\boldsymbol{w} - \boldsymbol{y}\|^2 \tag{4}$$

where $\boldsymbol{w}$ represents the amount invested in each asset. For each $i \in [n]$ denoting a timestep , the $i$-th row of $\boldsymbol{A}$ denotes the returns of the $d$ stocks at timestep $i$, and $y_i$ the return of the index fund. In our scenario, we seek to limit to a value $D > 0$ the amount of transactions in each of $c$ activity sector (group) of the portfolio (e.g. Industrials, Healthcare, etc.), denoted as $G_i$ for $i \in [c]$. We ensure such constraint through an $\ell_1$ norm constraint on each group: $\Gamma = \{\boldsymbol{w} \in \mathbb{R}^d : \forall i \in [c], \|\boldsymbol{w}_{G_i}\|_1 \leq D\}$, where $\boldsymbol{w}_{G_i}$ is the restriction of $\boldsymbol{w}$ to group $G_i$ (i.e. for $j \in [d]$, $\boldsymbol{w}_{G_{ij}} = \boldsymbol{w}_j$ if $j \in G_i$ and 0 otherwise). In our case, $\boldsymbol{y}$ denotes the daily returns of the S&P500 index from January 1, 2021, to December 31, 2022, and $\boldsymbol{A}$ the returns of the corresponding $d = 497$ assets (over $c = 11$ sectors) of the index during such period. We choose $k = 15$ and $D = 50$. We also apply our algorithms to additional financial indices (CSI300 and HSI) in Appendix F.1.

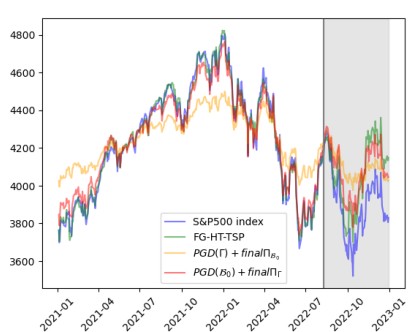

Figure 2: Index Tracking with Sector constraints

**Results.** Up to our knowledge, there are no closed form for the Euclidean projection onto $\mathcal{B}_0(k) \cap \Gamma$, but the two-step projection can easily be done by projecting onto the $\ell_1$ ball for each sector independently. We compare our algorithm (FG-HT-TSP) to two naive baselines: (a) the first one, called "PGD($\Gamma$) + final$\Pi_{\mathcal{B}_0}$", consists in only ensuring the constraints in $\Gamma$, followed at the end of training by a simple hard-thresholding step to keep the $k$ largest components of $\boldsymbol{w}$ in absolute value, and (b) the second one, called "PGD($\mathcal{B}_0$)+final$\Pi_\Gamma$", consists in running vanilla IHT, followed at the end of training by a simple projection onto $\Gamma$ to keep $\boldsymbol{w}$ in $\Gamma \cap \mathcal{B}_0$. We plot in Figure 2 the value of the returns for (i) the tracked index, (ii) our index (output of FG-HT-TSP), and (iii) our two baselines (a) and (b). We learn the weights of the portfolio on 80% of the considered period, and evaluate the out of sample (test set) performance on the remaining 20% (shaded area in the figure). As we can observe, the true index is successfully tracked by our method (FG-HT-TSP) (better than the two baselines as can be observed in particular on the train-set: the green curve is the one which is the closest to the blue one), and our algorithm solution spans 9 sectors, therefore it is well diversified, which illustrates the applicability of our method in practice.

## 6 CONCLUSION

In this paper, we provided global convergence guarantees for variants of Iterative Hard Thresholding which can handle extra convex constraints which are support-preserving, via a two-step projection algorithm. We provided our analysis in the deterministic, stochastic, and zeroth-order settings. To that end, we used a variant of the three-point lemma, adapted to such mixed constraints, which allowed to simplified existing proofs for vanilla constraints (and to provide a new kind of result in the ZO setting), as well as obtaining new proofs in such combined constraints setting. We illustrated the applicability of our algorithm on several sparse learning tasks. Finally, it would also be interesting to extend this work to a broader family of sparsity structures and constraints, for instance to matrices or graphs. We leave this for future work.

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
