# OpenReview forum: "Optimization over Sparse Restricted Convex Sets via Two Steps Projection"
_ICLR.cc/2024/Conference — Submitted to ICLR 2024_

### Official Review · Reviewer_cMSm · 2023-10-30

**Soundness:** 2 fair
**Presentation:** 2 fair
**Contribution:** 2 fair
**Rating:** 8
**Confidence:** 1

**Summary:**

Due to a medical emergency, I am unable to assess this article.

**Strengths:**

N/A

**Weaknesses:**

N/A

**Questions:**

N/A

---

> ### Author Response · Authors · 2023-11-22
> **Response to Reviewer cMSm**
>
> Dear Reviewer cMSm, we sincerely appreciate your support for our paper, even amid a challenging medical emergency.

---

### Official Review · Reviewer_LAEa · 2023-10-31

**Soundness:** 2 fair
**Presentation:** 3 good
**Contribution:** 3 good
**Rating:** 6
**Confidence:** 3

**Summary:**

The paper studies sparse optimizations problems where the underlying constraints consist of the intersection of an $\ell_0$ constraint and an extra constraint $\Gamma$. The authors introduce a IHT algorithm for optimizing over such sets, which uses a novel two step projection (TSP) procedure. By deriving an extension of the three point lemma for this setting, convergence guarantees are derived for this algorithm, assuming the objective satisfies the RSC and RSS conditions. The algorithm is extended to the stochastic and zeroth order settings where similar guarantees are derived.

**Strengths:**

1. The paper is written well overall, with a clear state-of-art, problem description, notation, and exposition.


2. The proposed TSP operator  is novel in my view, and also very natural.  The extension of the three point lemma derived in this paper is interesting, and is the main technical tool for obtaining the convergence guarantees.


3. The convergence results are interesting and I think these would be of interest to those working in the areas of sparse signal recovery.

**Weaknesses:**

1. At present there is no proof outline, which makes it difficult to understand the novelty in the proof technique compared to the literature. I understand there is a space constraint, but given that the main contributions of the paper are theoretical, this is relevant.

2. In the experiments, there is currently no comparison with existing methods. I think this would have been nice to illustrate on synthetic data, and would also provide an empirical validation of the theoretical results.

**Questions:**

I have the following minor remarks\questions.

1. I think the statement of assumption 3 is currently written in the form of a definition.

2. In Remark 1, the fourth bullet should ideally be the second bullet, just to introduce the notion of convex symmetric sets first.

3. In Algorithm 2, I am a bit confused about the notation $\mathcal{S}$. Shouldn't the index set $\mathcal{S}_t$ be a subset of $[n]$? Also in the summation index, it should be $i_t$ instead of $i$?

4. The convergence results pertain to the objective value, but can something be said for the convergence of the iterates to the global minimum? (provided there is a unique global minimum of course)

---

> ### Author Response · Authors · 2023-11-18
> **Response to Reviewer LAEa**
>
> Thank you for your insightful review and appreciation of our work.
>
> > **Your comment:** At present there is no proof outline, which makes it difficult to understand the novelty in the proof technique compared to the literature.
>
> **Our response:** Thank you for your suggestion, we have added a proof outline for Theorem 1 in the main paper, and additional explanations on the key parts of our proofs at the beginning of Appendix E.1 and E.3 (from the new revision).
>
> > **Your comment:**  In the experiments, there is currently no comparison with existing methods. I think this would have been nice to illustrate on synthetic data, and would also provide an empirical validation of the theoretical results.
>
> **Our response:** In our experiment from the main paper (Section 5) about the portfolio optimization problem, we compared our algorithm with a naive algorithm: a projected gradient descent with only $\Gamma$ as constraint, followed by a hard-thresholding at the last iteration. In the new revision, we also added an additional baseline which is IHT followed, after its last iteration, by Euclidean projection onto $\Gamma$.
> Indeed, with the constraint $\Gamma$ from this experiment, there is, up to our knowledge, no closed-form for the Euclidean projection (EP) onto our mixed constraint, which is why there are no other baselines to compare our algorithm with. Regarding more synthetic experiments to illustrate our theoretical results, we compared the TSP and EP in our section F.2 which also includes some discussions on our theoretical results and the role of $\rho$ (we have updated that section to match our new Assumption 3 in the new revision), and following your suggestion, we have added a new section in Appendix F.4 with more synthetic experiments to specifically discuss our novel bounds and their dependence on $\rho$.
>
> > **Your comment:** I think the statement of assumption 3 is currently written in the form of a definition.
>
> **Our response:** Thank you, we have rewritten all assumptions in an assumption form.
>
> > **Your comment:** In Remark 1, the fourth bullet should ideally be the second bullet, just to introduce the notion of convex symmetric sets first.
>
> **Our response:** Thank you, we have updated our Assumption on $\Gamma$ as described above per your suggestion, as well as the following Remark and this has fixed such problem.
>
> > **Your question:** Shouldn't the index set $S$ be a subset of $[n]$? Also in the summation index $i_t$, it should be instead of $i$?
>
> **Our response:**  Thank you, you are right, we have replaced $\mathcal{S}$ by $[n]$ and fixed the index $i$ into $i\_t$ in the new revision.
>
> > **Your question:** The convergence results pertain to the objective value, but can something be said for the convergence of the iterates to the global minimum? (provided there is a unique global minimum of course)
>
> **Our response:**  Yes, actually, convergence in terms of $\\|\boldsymbol{w} - \bar{\boldsymbol{w}} \\|$, with $\bar{\boldsymbol{w}} = \arg\min\_{\boldsymbol{v} \in \mathbb{R}^d ~ \text{s.t.} ~ \\| \boldsymbol{v} \\|\_0 \leq \bar{k}} R(\boldsymbol{v})$, in our extra constraints setting, can be immediately obtained from the result without extra constraint, for an even simpler algorithm which just projects onto $\Gamma$ at the last step. Indeed, suppose one runs vanilla IHT (or any variant such as the stochastic or zeroth-order one), without the extra projection step, to obtain a bound of the form $\\|\boldsymbol{\hat{w}\_T} - \bar{\boldsymbol{w}} \\|^2 \leq A\_T + B$ for some $A\_T$ and $B$, with $\boldsymbol{\hat{w}}\_T$ the output ($A\_T$ is usually a decreasing term, and $B$ a system error (in red in Table 1)). Then, one can simply consider $\boldsymbol{\hat{z}\_T} = \Pi\_{\Gamma}(\boldsymbol{\hat{w}\_T})$, i.e. the projection of the output onto $\Gamma$. Since projection onto a convex set is non-expansive, one will then have $\\|\boldsymbol{\hat{z}\_T} - \bar{\boldsymbol{w}} \\|^2 \leq \\|\boldsymbol{\hat{w}\_T} - \bar{\boldsymbol{w}} \\|^2 \leq A\_T + B$, and therefore the bound will remain unchanged. Note that this cannot be improved in general, since if it could be, one would take $\Gamma = \mathbb{R}^d$ and one would obtain better bounds even in the case without extra constraints $\Gamma$. This is why we did not consider convergence in terms of $\\|\boldsymbol{w} - \boldsymbol{\bar{w}} \\|$ in our paper: such result is somehow trivial as it does not require any analysis.  As an intuitive explanation, we could say that adding an extra constraint will restrict the space even further, therefore it is easy to see that iterates will only get closer to $\bar{\boldsymbol{w}}$: however, what is more interesting and needs to rethink the whole proof of convergence is the impact that such extra constraints have on the risk $R$: this is what we tackle in our paper.

---

> > ### Author Response · Authors · 2023-11-22
> > **Gratitude for your review**
> >
> > Dear Reviewer LAEa, thank you once more for your valuable review; should you have any remaining concerns, please don't hesitate to reach out.

---

> > ### Comment · Reviewer_LAEa · 2023-11-23
> >
> > Sorry for the delayed response. I have read the response of the authors to my queries and I am satisfied with the response. I will keep my original score for this paper.

---

> > > ### Author Response · Authors · 2023-11-23
> > > **Response to Reviewer LAEa**
> > >
> > > Dear reviewer LAEa, thank you for acknowledging our response, and confirming your appreciation of our work.

---

### Official Review · Reviewer_MFJV · 2023-11-04

**Soundness:** 2 fair
**Presentation:** 2 fair
**Contribution:** 2 fair
**Rating:** 3
**Confidence:** 4

**Summary:**

This paper studies rate of convergence of Iterative Hard Thresholding (IHT) on sparse regression with extra constraints on regression coefficients.

**Strengths:**

This paper studies rate of convergence of Iterative Hard Thresholding (IHT) on sparse regression with extra constraints on regression coefficients.

**Weaknesses:**

The work is very much similar to literature (Jain et al., 2014; Nguyen et al., 2017; Li et al., 2016; Shen & Li, 2017; de Vazelhes et al., 2022). The referee is afraid that adding an extra constraint \Gamma of the regression coefficients making any fundamental difference. More importantly, it does not make much sense to enforce an extra constraint \Gamma. The solution only converges to stationary points. And the authors never verify the assumptions with a practical example.

**Questions:**

The work is very much similar to literature (Jain et al., 2014; Nguyen et al., 2017; Li et al., 2016; Shen & Li, 2017; de Vazelhes et al., 2022). The referee is afraid that adding an extra constraint \Gamma of the regression coefficients making any fundamental difference. More importantly, it does not make much sense to enforce an extra constraint \Gamma. The solution only converges to stationary points. And the authors never verify the assumptions with a practical example.

---

> ### Author Response · Authors · 2023-11-18
> **Response to Reviewer MFJV [1/3]**
>
> Thank you for your insightful review. We sincerely hope that the main concerns raised in the review can be clarified by the following responses.
>
> > **Your comment:**  The work is very much similar to literature (Jain et al., 2014; Nguyen et al., 2017; Li et al., 2016; Shen \& Li, 2017; de Vazelhes et al., 2022).
>
> **Our response:** As we elaborate upon in the global rebuttal, our work adds some significant contributions to the previous works in IHT:
>
>   1. Even in the case of no extra constraints $\Gamma$, our developed three-point lemma allows to greatly simplify the previously existing proofs of Jain et al. (2014) (Proof of Theorem 1, Appendix B.1), and Zhou et al. (2018) (Proof of Theorem 2, Appendix B.3): those proofs use intricate observations on the support sets of the iterates, which as such are hardly applicable and extendable. But our corresponding proofs (Proof of Theorem 4 followed by Proof of Theorem 1, and Proof of Theorem 5 followed by Proof of Theorem 2, and finally Proof of Theorem 6 followed by Proof of Theorem 3), are much simpler using more standard tools, which are similar to the ones from convex optimization (three-points lemma, telescopic sums etc), and as such are more extensible to future settings. As an example of such power of extensibility, even in the case of no extra $\Gamma$, in addition to simplifying and improving existing proofs, we can also provide a new result in the zeroth-order case (Theorem 6, cf. Table 1, described in more details in our Appendix E.3.2): such result is a significant improvement over the result of de Vazelhes et. al. (2022), since it provides a convergence in risk $R(\boldsymbol{w}\_t)$, without system error (i.e. without the term in red in Table 1). In the IHT litterature, such type of results are usually harder to obtain than results in terms of $\\| \boldsymbol{w} - \bar{\boldsymbol{w}} \\|$, which also exhibit an impractical system error.
>
>   2. In terms of technical challenge, since we use a new tool for the proof (the three-points lemma), our work have necessitated to fully rethink the proofs of those variants of IHT, which therefore we believe is a significant contribution to the IHT literature.
>
>   3. And finally, as a main example of the power of our framework, and our main contribution, we provide the novel convergence results that we do in the case of extra constraint $\Gamma$.
>
> > **Your comment:** The referee is afraid that adding an extra constraint $\Gamma$ of the regression coefficients making any fundamental difference.
>
>
> **Our response:** First, before addressing your concern regarding extra constraints $\Gamma$, we would like to mention, as described above, that our work already provides significant contributions to the literature even in the case where there is no extra constraint. With that in mind, when we consider the case with extra constraints and try to build upon our previously introduced techniques, an important new difficulty arises, since the simpler version of our three-point lemma (i.e. Lemma 3 in Appendix) is not valid anymore (it is valid only for a vanilla $\ell\_0$ constraint) : this lead us to derive our Lemma 1, which is a three-point lemma which can take into account the extra constraints. This new lemma is key to our analysis and combined with a novel analysis which introduces the trade-off variable $\rho$, leads to our new results. Additionally, independently of our three-point lemma technique, even trying to directly extend the previous original proofs of convergence for IHT, for convergence in risk without system error, i.e. as in Jain et. al (2014), and Zhou et. al. (2018), in the case of extra constraints, would pose a challenge  (for convergence in terms of $\\|\boldsymbol{w}-\boldsymbol{\bar{w}}\\|$ it would indeed not change much, as mentioned in the response to Reviewer LAEA Q.4, which is why we focus on convergence in risk). Indeed, the proof of Jain et. al (2014) and Zhou et. al. (2018) crucially rely on a specific lemma (Lemma 1 in Jain et al. (2014), called Lemma 2 in Zhou et. al. (2018)), which does not hold anymore in the case of extra constraint. More precisely, such lemma states that, for a $\bar{k}$-sparse $\bar{\boldsymbol{w}}\in \mathbb{R}^d$ and for any $\boldsymbol{w} \in \mathbb{R}^d$:  $\\|\mathcal{H}\_k(\boldsymbol{w}) - \boldsymbol{w} \\|^2 \leq \frac{d-k}{d-\bar{k}} \\| \bar{\boldsymbol{w}} - \boldsymbol{w}||^2$. However, if we denote our mixed constraints set by $M$ ($= \mathcal{B}\_0 \cap \Gamma$ ), it does not hold in general that, for a $\bar{k}$-sparse $\bar{\boldsymbol{w}} \in M: \\|\Pi\_{M}(\boldsymbol{w}) - \boldsymbol{w} \\|^2 \leq \frac{d-k}{d-\bar{k}} \\| \bar{\boldsymbol{w}} - \boldsymbol{w}\\|^2$ (one can see this for instance in the case where $d=3$, $\bar{k} =1$,  $k = 2$, $\Gamma$ is the $\ell\_{\infty}$ unit ball, $\boldsymbol{w} = (10, 10, 0)$ and $\boldsymbol{\bar{w}} = (1, 0, 0)$). **[...Continued below...]**

---

> > ### Author Response · Authors · 2023-11-18
> > **Response to Reviewer MFJV [2/3]**
> >
> > **[...Continuing from above...]** In other words, the proofs of Jain et. al (2014) and Zhou et. al (2018) rely on the fact that taking the hard-thresholding of a vector does not change that vector significantly (as it just removes its smallest components in magnitude): but this is not true anymore in general for the projection onto mixed constraints $\Gamma \cap \mathcal{B}\_0(k)$ (cf. our aforementioned example). As such, adding extra constraints imposes to fully rethink the proofs for convergence in risk without system error.  Finally, the discussion above considered the difficulty of adding extra constraints $\Gamma$ from the perspective of the hard-thresholding literature. As a complementary view, from the perspective of convex constrained optimization with a constraint $\Gamma$, the difficulty of adding an extra $\ell\_0$ constraint comes from the fact that, since the resulting mixed constraint $\Gamma \cap \mathcal{B}\_0$ is *not convex anymore*, the *convex* three-point lemma (which relates the distances between $\boldsymbol{w} \in \mathbb{R}^d$, $\Pi\_\Gamma(\boldsymbol{w})$, and $\boldsymbol{\bar{w}} \in \Gamma$ in a specific way) does not hold anymore. Trying to stay as close as possible to the (simpler) convex optimization proofs is also what motivated us to introduce our new non-convex $\ell\_0$ three points lemma with extra constraints (Lemma 1).
> >
> >
> >
> > > **Your comment:** More importantly, it does not make much sense to enforce an extra constraint $\Gamma$.
> >
> > **Our response:** Enforcing a constant $\Gamma$ is indeed necessary in a variety of practical use cases, some of which we have mentioned in Introduction as well as Experiments: a prominent example is the case of index tracking or portfolio optimization, also mentioned for instance in Lu et al. (2015) as well as [3], [4], [5]: in a typical setting, in addition to the cardinality constraints (which enforces that one invests in only $k$ assets), one may have a maximum budget to invest in the portfolio, which can be enforced through an additional $\ell\_1$ norm constraint (if shorts are allowed), or an extra $\ell\_1$-norm constraint combined with a positivity constraint, for instance (if no shorts are allowed). Additional constraints such as box constraints (to ensure that a maximum amount of investment per asset is not exceeded), or sector constraints (to ensure that the portfolio is diversified, as we illustrate in our experiments) can also be enforced. As another example, positivity constraints can also be enforced in addition to cardinality constraints in the case of sparse non-negative factorization, for estimating the hidden components, see for instance [6]. Lastly, extra constraints such as box constraints are often enforced in sparse adversarial attacks to render those attacks imperceptible, in the white-box [6] and black-box [7] settings. We thank you for highlighting that point and have updated the new revision accordingly.
> >
> > > **Your comment:** The solution only converges to stationary points.
> >
> > **Our response:**  Actually, the aim of our paper is to prove global convergence guarantees, not local convergence guarantees like convergence to stationary points. Indeed, so far, local convergence guarantees, more precisely, guarantee of convergence to a stationary point, already exist in the literature of IHT with extra constraints, with the works of Lu et al, and Beck et al, as can be seen in Table 1: however, global convergence guarantees under RSC and RSS assumptions, like the ones in (Jain et al.), (Zhou et al., Li et al., Vazelhes et al. etc), *but in the case of extra constraints*, were still missing: this is precisely the gap that we addresss in the paper, and we do so in a principled way, which, as mentioned above, also improves as a byproduct the previous analyzes of IHT in the vanilla setting without extra constraints.
> >
> >
> >
> > > **Your comment:** And the authors never verify the assumptions with a practical example.
> >
> > **Our response:** Actually, we chose our experiments specifically to illustrate real-life tasks where the assumptions of our theorems are verified (except for the adversarial attacks experiments, which is given only for indicative purposes). Indeed, one can easily check that they verify the assumptions in the paper, as described below (we have added such explanations in the new revision): **[...Continued below...]**

---

> > > ### Author Response · Authors · 2023-11-18
> > > **Response to Reviewer MFJV [3/3]**
> > >
> > > **[...Continuing from above...]**
> > >
> > > - For the index tracking problem (Section 5 and Appendix F.1 from the new revision):
> > >
> > >      - **Assumption 1** is verified since the cost function is quadratic, with a design matrix of size $n > d$ (except in the case of S\&P500). As can be expected with such matrices in general, the Hessian $\boldsymbol{H} = 2 \boldsymbol{A}^{\top} \boldsymbol{A}$ is positive-definite (we have indeed verified in our code that it is). Therefore the RSC constant is bounded below by $\lambda\_{\min}$ where $\lambda\_{\min}$ is the smallest eigenvalue of $2 \boldsymbol{A}^{\top} \boldsymbol{A}$. Note that for S\&P500, strong convexity is not verified since $d > n$: however, since we take $k=15$, with high-probability (i.e. unless we can find $s=2k=30$ columns of $\boldsymbol{A}$ that are exactly linearly dependent), RSC should be verified.
> > >
> > >      - **Assumption 2 and Assumption 5** are both verified since the cost function is quadratic, therefore the (strong) RSS constant is bounded above by $2 \\| \boldsymbol{A}\\|\_s^2$, where $\\| \cdot \\|\_s$ denotes the spectral norm.
> > >
> > >      - **Assumption 3** is verified since projection onto $\Gamma$ can be done group-wise  (i.e. sector-wise), and for each group the projection is onto an $\ell\_1$ ball, which is a convex sign-free set (which is support-preserving from Remark 1), therefore, overall, $\Gamma$ is support-preserving).
> > >
> > > - For multiclass logistic regression: (Section F.3 from the new revision)
> > >
> > >
> > >      - **Assumption 1** is verified thanks to the added squared $\ell\_2$ regularization, which makes the problem strongly convex and hence also restricted strongly convex.
> > >
> > >      - **Assumption 2 and Assumption 5** are both verified since the problem is smooth with a constant $L$ as described in Equation (64) from Appendix F.3 (in our new revision), and therefore such constant is also a valid (strong) restricted-smoothness constant.
> > >
> > >      - **Assumption 3** is verified since, similarly as in the index tracking experiments from Section 5, projection onto $\Gamma$ can be done group-wise, and for each group, the projection is onto an $\ell\_1$ ball, which is a convex sign-free set (which is support-preserving from Remark 1), therefore, overall, $\Gamma$ is support-preserving.
> > >
> > >
> > > *[2] Simultaneous pursuit of out-of-sample performance and sparsity in index tracking portfolios. A. Takeda et al., Computational Management Science 10(1) 21–49, 2012.*
> > >
> > > *[3] An efficient optimization approach for a cardinality constrained Index tracking problem. F. Xu et al., Optim. Method Softw., 2015.*
> > >
> > > *[4] On the minimization over sparse symmetric sets: projections, optimality conditions and algorithms.  A. Beck and N. Hallak.  Mathematics of Operations Research, 2015.*
> > >
> > > *[5] Non-negative sparse coding, P. Hoyer, Proceedings of the 12th IEEE workshop on neural networks for signal processing, 2002.*
> > >
> > > *[6] Sparse and imperceivable adversarial attacks. Croce F., ICCV, 2019.*
> > >
> > > *[7] Sparse and imperceptible adversarial attack via a homotopy algorithm. Zhu M., ICML, 2021.*

---

> > > > ### Author Response · Authors · 2023-11-22
> > > > **Gratitude for your review**
> > > >
> > > > Dear Reviewer MFJV, thank you once more for your valuable review; should you have any remaining concerns, please don't hesitate to reach out.

---

### Official Review · Reviewer_r73h · 2023-11-05

**Soundness:** 3 good
**Presentation:** 2 fair
**Contribution:** 2 fair
**Rating:** 3
**Confidence:** 4

**Summary:**

The manuscript investigated a constrained optimization setting where the goal is to minimize an objective function while satisfying some sparsity constraints. A two-step-projected gradient-based approach is proposed. Their performance is analyzed for both stochastic and non-stochastic settings.

**Strengths:**

Understanding optimization with sparsity constraint is a long-standing open question. This submission is one of not many works that formulates and provides analysis towards that direction.

**Weaknesses:**

Technically, there are hidden constraints to the main results that could significantly limit the strength of the proposed results. For example, Theorem 1 and the remarks that follow imply that the proposed two-step projection guarantees the convergence to the global minimum if the minimizer is sparse, which is not true in the most generic case (e.g., when k=1). However, the proof was made possible due to the requirement of $k\geq \frac{4(1-\rho)^2 L_s^2}{\rho^2 \nu_s^2} \overline{k}$, which in fact requires the sparsity constrain has to be weak, i.e., lower bounded by the square of the condition number for the non-trivial case of $\overline{k}\neq 0$. So, no guarantee is provided for the important case of fixed sparsity $k$, even if we allow k to be an arbitrarily large constant.

Some other minor comments and suggestions:

1. The main results are rather hard to read due to the fact that many needed notations are defined informally inline in different sections, for example, H_k in Theorem 1. Possible ways of improvement include either defining the notations formally in a definition environment or naming the notation so they can be easily searched.

2. Interpretation of  $\rho$ in the main result: We are generally interested in the achievable result of $R(w_t)$ and the inequality presented in Theorem 1 seems to be a more intermediate result where the role of parameter $\rho$ could be unclear to the readers. For readability, the reviewer suggests that Remarks 3 and 4 should be summarized as a main theorem, then Theorem 1 is presented as a technical result for proving them.

**Questions:**

As the presented theorem requires a sparsity constraint k that grows unbounded with respect to the condition number. Would it be possible to modify the analysis or the algorithm to extend the results for any large constant k?

---

> ### Author Response · Authors · 2023-11-18
> **Response to Reviewer r73h**
>
> Thank you for your insightful review. We sincerely hope that the main concerns raised in the review can be clarified by the following responses.
>
> > **Your comment:** Technically, there are hidden constraints to the main results that could significantly limit the strength of the proposed results. ...So, no guarantee is provided for the important case of fixed sparsity $k$, even if we allow k to be an arbitrarily large constant.
>
>
> **Our response:** Thank you for your suggestions. Regarding the requirement $k \geq \frac{1(1 - \rho)^2 L\_s^2}{\rho^2 \nu\_s^2}  \bar{k}$, we would like to highlight that such a relaxation of $k$, depending on the condition number $\frac{L\_s}{\nu\_s}$ is present in all the literature of hard-thresholding in the RSC-RSS setting, as can be shown by the references in our Table 1 (column "$k$") (namely, Jain et al. (2014), Nguyen et al. (2017), Li et al. (2016), Zhou et al. (2018), and de Vazelhes et al. (2022)). This stems from the fact that the problem is NP-hard, and even hard to approximate [1] and as such the original problem $\min\_{\boldsymbol{w}} f(\boldsymbol{w}) ~ \text{s.t.} ~ \\| \boldsymbol{w} \\|\_0 \leq k$ cannot be solved in general to global optimality in its original form with a fixed $k$. In fact, it was actually shown in [1] that the $\kappa^2 = \frac{L\_s^2}{\nu\_s^2}$ factor of the sparsity relaxation cannot be improved in the analysis of IHT (Iterative Hard Thresholding). Finally, we highlight that usually hard-thresholding methods are used for very high-dimensional problems, and as such, even relaxing the sparsity by a constant factor provides sparse enough solutions.
>
>
> > **Your comment:** The main results are rather hard to read due to the fact that many needed notations are defined informally inline in different sections... so they can be easily searched.
>
> **Our response:** Thank you for your suggestion, we have gathered our notations at the beginning of Section 2 (Preliminaries), and have added a section in the supplemental referencing them, for sake of clarity.
>
> > **Your comment:** For readability, the reviewer suggests that Remarks 3 and 4 should be summarized as a main theorem, then Theorem 1 is presented as a technical result for proving them.
>
> **Our response:**  Thank you for your suggestion. We agree that the result in terms of $R(\boldsymbol{w}\_t)$ is indeed clearer: we have rewritten Theorem 1 so as to display the result in $R(\boldsymbol{w}\_t)$ as per Remark 3 and 4 in the new revision.
>
> > **Your question:** Would it be possible to modify the analysis or the algorithm to extend the results for any large constant k?
>
> **Our response:** In general, it would not be possible to modify the analysis to extend the result for any large constant $k$, since it is known that $k$ must be in $O(\kappa^2 \bar{k})$, where $\kappa$ is the condition number, and such dependence is not improvable in the analysis of IHT (as described in [1]). This fundamental bottleneck is essentially due to the NP-hardness of such a non-convex optimization problem.
>
>
> [1] Iterative hard thresholding with adaptive regularization: Sparser solutions without sacrificing runtime, K. Axiotis and M. Sviridenko, ICML, 2022

---

> > ### Author Response · Authors · 2023-11-22
> > **Gratitude for your review**
> >
> > Dear Reviewer r73h, thank you once more for your valuable review; should you have any remaining concerns, please don't hesitate to reach out.

---

### Author Response · Authors · 2023-11-18
**Global response**

We would like to thank all the reviewers for their insightful comments and useful suggestions. The main concern revolves around the value of our two-step projected IHT algorithm and theory added beyond the prior IHT literature, which we sincerely hope can be addressed by the following brief summary of the main contributions of our paper:

1. First, our work provides significant contributions to the IHT literature *even in the case where no additional constraints are present*: we have achieved so thanks to a new three-point lemma adapted to projection onto the $\ell\_0$ pseudo-ball, which allows to make proofs of IHT for convergence in risk (i.e. objective value), without system error, much closer to the standard framework of convex optimization proofs (even if our problem is non-convex). This is significant since, in the IHT literature, such kind of proofs are harder to obtain than proofs in terms of estimation error $\\|\boldsymbol{w} - \bar{\boldsymbol{w}} \\|$. More precisely, as an illustration of this, one can compare the proof of our Theorem 4 and 5 in Appendix, to, respectively, the proofs in Jain et al. (2014) (Proof of Theorem 1, Appendix B.1), and Zhou et al. (2018) (Proof of Theorem 2, Appendix B.3): such proofs use intricate considerations about the support sets of the iterates and gradients, which we can avoid in our proof framework. As such, we believe our proof framework will be helpful in the literature to extend  more easily IHT proofs (for convergence in risk without system error) to a variety of settings. Finally, *still in such setting without additional constraints*, our framework allows us to provide a *new result*: *the first proof of convergence in risk without system error for a zeroth-order hard-thresholding method*: our Theorem 6, which is a significant improvement over the work of de Vazelhes et. al. (2022) which only tackled convergence in terms of $\\|\boldsymbol{w} - \bar{\boldsymbol{w}} \\|$, and exhibit a system error (in red in our Table 1).

2. Then, as another demonstration for the extensibility of our framework, we extend such a three-point lemma to the case where additional constraints are present in the problem (our main setting of interest, Problem (1)). Such extra constraints $\Gamma$ are indeed necessary in a variety of practical use cases, such as (i) sparse portfolio optimization with maximum total or asset-wise budget constraints with or without shorts allowed, (ii) non-negative sparse matrix factorization, and (iii) (white box and black box) sparse adversarial attacks under norm constraints (cf. answer to Reviewer MFJV for more details on such use cases). In such framework, we use a new proof technique tied to the aforementioned three points lemma (extended to the case with extra constraints), which allows us to tackle this new setting for which up to now there was no global convergence guarantees, as can be seen in our Table 1. Our new results also exhibit a novel compromise between the sparsity of the iterates and the objective value, as measured by a parameter $\rho$, and which we discuss in the paper.

We have carefully revised the manuscript based on the reviews. The following is a summary of major changes (we have colored our modifications in red in the revised documents for convenience):

1. We made a slight modification in our Assumption 3: the sets that we consider in fact need to be "support preserving", not "restricted convex": this is only needed to fix a slight technical issue that we had in the proof of Lemma 1 (which is now fixed), and the rest of the proofs and experimental results is not impacted. Only one section in Appendix (previously located in section E.2) was impacted because the simplex constraint that we considered there does not verify our updated Assumption 3, so we have updated it, but this does not impact the rest of the conclusions in the paper.

2. We have added an additional baseline in our index tracking example: IHT followed by a projection onto $\Gamma$ for the last step: this further illustrates the applicability of our two-step projection method (which also performs better than this new baseline), following reviewer LAEa's suggestions.

3. We added a new section in the Appendix, with a synthetic example to illustrate the sparsity/objective value compromise, following reviewer LAEa's suggestions.

4. We added several modifications in the text of the revision: we rewrote our Theorems in the simpler form of their following Remarks, we added proof sketches, we detailed why our experiments verify assumptions 1 to 3 in practice, clarified the notations, and fixed some typos and minor changes, following the suggestions of all the reviewers.

We hope that the given concerns have been addressed satisfactorily in the revised version, and in the following point-by-point responses to the reviewers' comments as well.

---

### Meta-Review · Area_Chair_dggy · 2023-12-11

**Metareview:**

Sparsity have already been well studied in the field of Machine learning.
The paper investigates Iterative hard thresholding type(IHT) algorithms when the constraint in not only the l_0 norm but also additional convex constraints.
The contributions of the paper are as follows.
It extends the three point lemma which has many uses including simplifying existing proofs.
It provides IHT style new algorithm with global convergence guarantees.
The authors have made several changes to the manuscript in response to the rebuttals  which includes changing the title to include the phrase "support preserving".  I think the amended paper needs more extensive review.
After considering the reviews, rebuttal, reading the original and revised paper, it will hard to argue for
acceptance at a extremely competitive fora such as ICLR.

**Justification For Why Not Higher Score:**

Given the large body of work on sparsity the paper needs to have more extensive discussion to position the contributions better.
The revised version of the paper is substantially different from the original draft. It needs an extensive review.

**Justification For Why Not Lower Score:**

N/A

---

### Decision · Program_Chairs · 2024-01-16

Reject